# Spatio–Temporal Hilbert Maps for Continuous Occupancy Representation in Dynamic Environments

**Ransalu Senanayake**
University of Sydney
rsen4557@uni.sydney.edu.au

**Lionel Ott**
University of Sydney
lionel.ott@sydney.edu.au

**Simon O'Callaghan**
Data61/CSIRO, Australia
simon.ocallaghan@data61.csiro.au

**Fabio Ramos**
University of Sydney
fabio.ramos@sydney.edu.au

## Abstract

We consider the problem of building continuous occupancy representations in dynamic environments for robotics applications. The problem has hardly been discussed previously due to the complexity of patterns in urban environments, which have both spatial and temporal dependencies. We address the problem as learning a kernel classifier on an efficient feature space. The key novelty of our approach is the incorporation of variations in the time domain into the spatial domain. We propose a method to propagate motion uncertainty into the kernel using a hierarchical model. The main benefit of this approach is that it can directly predict the occupancy state of the map in the future from past observations, being a valuable tool for robot trajectory planning under uncertainty. Our approach preserves the main computational benefits of static Hilbert maps — using stochastic gradient descent for fast optimization of model parameters and incremental updates as new data are captured. Experiments conducted in road intersections of an urban environment demonstrated that spatio-temporal Hilbert maps can accurately model changes in the map while outperforming other techniques on various aspects.

## 1 Introduction

We are in the climax of driverless vehicles research where the perception and learning are no longer trivial problems due to the transition from controlled test environments to real world complex interactions with other road users. Online mapping environments is vital for action planing. In such applications, the state of the observed world with respect to the vehicle changes over time, making modeling and predicting into the future challenging. Despite this, there is a plethora of mapping techniques for static environments but only very few instances of truly dynamic mapping methods. Most existing techniques merely consider a static representation, and as parallel processes, initialize target trackers for the dynamic objects in the scene, updating the map with new information. This approach can be effective from a computational point of view, but it disregards crucial relationships between time and space. By treating the dynamics as a separate problem from the space representation, such methods cannot perform higher level inference tasks such as what are the most likely regions of the environment to be occupied in the future, or when and where a dynamic object is most likely to appear.

In occupancy grid maps (GM) [1], the space is divided into a fixed number of non-overlapping cells and the likelihood of occupancy for each individual cell is estimated independently based on sensor measurements. Considering the main drawbacks of the GM, discretization of the world and disregarding spatial relationship among cells, Gaussian process occupancy map (GPOM) [2] enabled

continuous probabilistic representation. In spite of its profound formulation, it is less pragmatic for online learning due to $\mathcal{O}(N^3)$ computational cost in both learning and inference, where $N$ is the number of data points. Recently, as an alternative, static Hilbert maps (SHMs) [3, 4] was proposed, borrowing the two main advantages of GPOMs but at a much lower computational cost. As a parametric technique, SHMs have a constant cost for updating the model with new observations. Additionally, the parameters can be learned using stochastic gradient descent (SGD) which made it computationally attractive and capable of handling large datasets. Nonetheless, all these techniques assume a static environment.

Although attempts to adapt occupancy grid maps to dynamic environments and identify periodic patterns exist [5], to the best of our knowledge, only dynamic Gaussian processes occupancy maps (DGPOM) [6] can model occupancy in dynamic environments in a continuous fashion. There, velocity estimates are linearly added to the inputs of the GP kernel. This approach, similar to the proposed method, can make occupancy predictions into the future. However, being a non-parametric model, the cost of inverting the covariance matrix in DGPOM grows over time and hence the model cannot be run in real-time.

In this paper, we propose a method for building continuous spatio-temporal Hilbert maps (STHM) using "hinged" features. This method builds on the main ideas behind SHM and generalize it to dynamic environments. To this end, we formulate a novel methodology to permeate the variability in the temporal domain into the spatial domain, rather than considering time merely as another dimension. This approach can be used to predict the occupancy state of the world, interpolating not only in space but also in time. The representation is demonstrated in highly dynamic urban environments of busy intersections with cars moving and turning in both directions obeying traffic lights. In Section 2, we lay the foundation by introducing SHMs and then, we discuss the proposed method in Section 3, followed by experiments and discussions in Section 4.

## 2   Static Hilbert maps (SHMs)

A static Hilbert map (SHM) [3] is a continuous probabilistic occupancy representation of the space, given a collection of range sensor measurements. As in almost all autonomous vehicles, we assume a training dataset consisting of locations with associated occupancy information obtained from a range sensor — in the case of a laser scanner (i.e. LIDAR), points along the beam are unoccupied while the end point is occupied — the model predicts the occupancy state of different locations given by query points.

**The SHM model:** Formally, let the training dataset be defined as $\mathcal{D} = \{\mathbf{x}_i, y_i\}_{i=1}^N$ with $\mathbf{x}_i \in \mathbb{R}^D$ being a point in $2D$ or $3D$ space, and $y_i \in \{-1, +1\}$ the associated occupancy status. SHM predicts the probability of occupancy for a new point $\mathbf{x}_*$ calculated as $p(y_*|\mathbf{x}_*, \mathbf{w}, \mathcal{D})$, given a set of parameters $\mathbf{w}$ and the dataset $\mathcal{D}$. This discriminative model takes the form of a logistic regression classifier with an elastic net regularizer operating on basis functions mapping the point coordinates to a Hilbert space defined by a kernel $k(\mathbf{x}, \mathbf{x}') : \mathcal{X} \times \mathcal{X} \to \mathbb{R}$ where $\mathbf{x}, \mathbf{x}' \in \mathcal{X} = \{\text{location}\}$. This is equivalent to kernel logistic regression [7] which is known to be computationally expensive due to the need of computing the kernel matrix between all points in the dataset. The crucial insight to make the method computationally efficient is to first approximate the kernel by a dot product of basis functions such that $k(\mathbf{x}, \mathbf{x}') \approx \Phi(\mathbf{x})^\top \Phi(\mathbf{x}')$. This can be done using the random kitchen sinks procedure [8, 9] or by directly defining efficient basis functions. Note that, [3] assumes a *linear machine* $\mathbf{w}^\top \Phi(\mathbf{x})$. Learning $\mathbf{w}$ is done by minimizing the regularized negative-log-likelihood using stochastic gradient descent (SGD) [10]. The probability that a query point $\mathbf{x}_*$ is not occupied is given by $p(y_* = -1|\mathbf{x}_*, \mathbf{w}, \mathcal{D}) = \left(1 + \exp(\mathbf{w}^\top \Phi(\mathbf{x}_*))\right)^{-1}$, while the probability of being occupied is given by $p(y_* = +1|\mathbf{x}_*, \mathbf{w}, \mathcal{D}) = 1 - p(y_* = -1|\mathbf{x}_*, \mathbf{w}, \mathcal{D})$.

## 3   Spatio-temporal hinged features (HF-STHM)

In this section, SHMs are generalized into the spatio-temporal domain. Though augmenting the inputs of the SHM kernel $\mathcal{X} = \{\text{location}\}$ as $\mathcal{X} = \{(\text{location}, \text{time})\}$ or $\mathcal{X} = \{(\text{location}, \text{time}, \text{velocity})\}$ is the naive method to build instantaneous maps, they cannot be used for predicting into the future mainly because they are not cable of capturing complex and miscellaneous spatio-temporal dependencies. As discussed in Section 3.3, in our approach, the uncertainty of

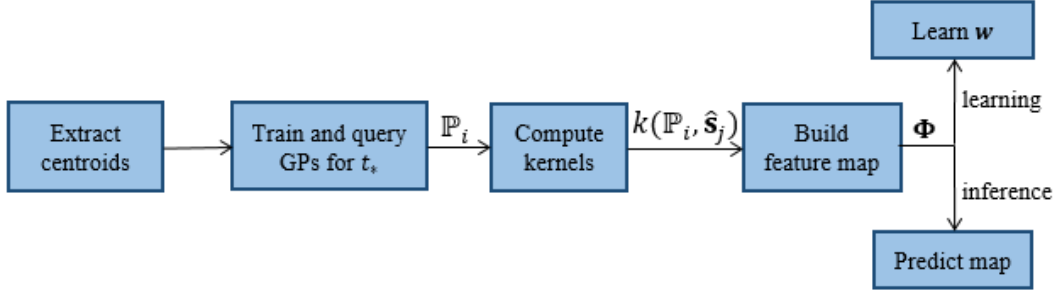

Figure 1: Motion centroids are collected over time from raw data (Section 3.1) and individual GPs are trained (input: centroids, output: motion information) to learn GP-hyperparameters (Section 3.2 and Figure 4). Then, the motion of data points at time $t_*$ ($= 0$ for present, $> 0$ for future, $< 0$ for past) are queried using the trained GPs and this motion distribution is fed into the kernel (Section 3.3). This implicitly embeds motion information into the spatial observations. Then a kernelized logistic regression model $\text{logistic}(\mathbf{w}^\top \Phi)$ is trained to learn $\mathbf{w}$. For a new query point in space, $\Phi(\text{longitude}, \text{latitude})$ is calculated using Equation 6 followed by $\text{sigmoidal}(\mathbf{w}^\top \Phi)$ to obtain the occupancy probability. These steps are repeated for each new laser scan.

dynamic objects is incorporated into the map. This uncertainty is estimated using an underlying Gaussian process (GP) regression model described in Section 3.2. The inputs for the GP are obtained using a further underlying model based on motion cluster data association which is discussed in Section 3.1. This way, locations are no more deterministic but each location has a probability distribution and hence the kernel inputs become $\mathcal{X} = \{\text{mean and variance of location}\}$. Sections 3.1–3.3 explain this three-step hierarchical framework in the bottom-to-top approach which are executed sequentially as new data are received. The method is summarized in Figure 1.

**Assumptions:** WOLOG, we assume that the sensor is not moving; the general case where the sensor moves is trivial if the motion of the platform is known. From a robotics perspective, we treat localization as a separate process and assume it is given for the purpose of introducing the method.

**Notation:** In this section, unless otherwise stated, input $\mathbf{x} = (x, y, t)$ are the longitude, latitude and time components and, $\mathbf{s} = (x, y)$ are merely the spatial coordinates. A motion vector (displacement) is denoted by $\mathbf{v} = (v_x, v_y)$, where $v_x$ and $v_y$ are the motion in $x$ and $y$ directions, respectively. A motion field is a mapping from space and time to a motion vector, $(x, y, t) \mapsto (v_x, v_y)$.

## 3.1 Motion observations

As the first step, motion observations are extracted from laser scans. Due to occlusions and sensor noise, extracting dynamic parts of a scene is not straightforward. Similarly, as the shapes of observed objects change over time (because the only measurement in laser is depth), morphology based object tracking algorithms and optical flow [11, 12] which are commonly used in computer vision are unsuitable. Therefore, we devise as a method that is robust to occlusions and noise without relying on the shape of the objects present in the scene. To obtain motion observations, taking raw laser scans as inputs and output motion vectors, the following two steps are performed.

### 3.1.1 Computing centroids of dynamic objects

As shown in Figure 2, firstly, a SHM is built from the raw scanner data at time $t$ and then it is binarized to produce a grid map containing occupied and free cells. Based on this grid map, observable areas where dynamic objects can appear are extracted. Next, dynamic objects are obtained by performing logical conjunction between an adaptive binary mask and the raw laser data. The final step is the computation of the centroid for each of these components.

### 3.1.2 Associating centroids of consecutive frames

Having obtained $N$ centroids for frame $t$ and, $M$ centroids for frame $t - 1$ from the previous step, we formulate the centroid–association as the integer program in Equation 1.

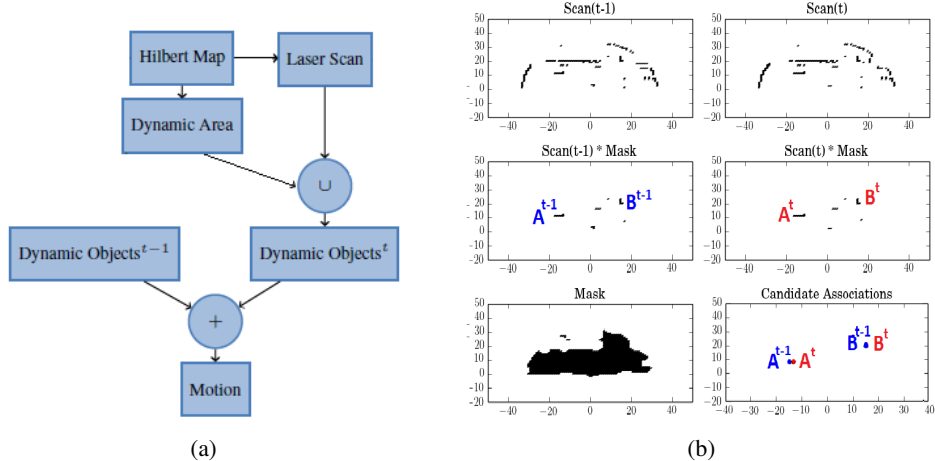

(a)                                   (b)

Figure 2: The various steps involved in computing motion observations discussed in Section 3.1 is shown in (a). The mask (lower left of (b)) is generated by applying morphological operations to the raw scans (top row). Taking the intersection between the mask and a raw scan yields the potential dynamic objects in a scene at a given time (middle row). The final centroid association of such connected components across two consecutive frames is shown in the bottom right frame.

$$\text{minimize} \quad \sum_{i=1}^{M} \sum_{j=1}^{N} d_{ij} a_{ij} \tag{1a}$$

$$\text{subject to} \quad \sum_{i=1}^{M} a_{ij} = 1, \quad j = 1, \ldots, N \tag{1b}$$

$$\sum_{j=1}^{N} a_{ij} = 1, \quad i = 1, \ldots, M \tag{1c}$$

$$a_{ij} \in \{0, 1\}, \tag{1d}$$

where $d_{ij}$ is the Euclidean distance between two centroids and $a_{ij}$ are the elements of the assignment matrix. In order to obtain valid assignment solutions $a_{ij}$, we impose that only one centroid from frame $t$ can be assigned to one centroid in frame $t - 1$, Equation 1b, and the vice versa with Equation 1c. Finally, we only allow integer solutions, Equation 1d. The solution to the above problem is obtained using the Hungarian method [13]. The asymptotically cubic computational complexity does not thwart online learning as the number of vehicles in the field of vision is typically very low (say, $< 10$). This forms the basis for obtaining the motion field which is described in the next section.

## 3.2 Motion prediction using Gaussian process regression

In this section we describe the construction of a model to predict the motion field as a mapping $(x, y, t) \rightarrow (v_x, v_y)$. We adopt a Bayesian approach that can provide uncertainty of a query point with a little amount of data. A Gaussian process (GP) regression model is instantiated for each new moving object and motion observations are collected over time until the object disappears from the robot's view. Each GP model has a different number of data points which grows over time during its lifespan. Nevertheless, this stage does not suffer from $\mathcal{O}(N^3)$ asymptotic cost of GPs because objects appear and disappear from the mapped area (say, the number of GPs $< 20$ and $N < 50$ for each GP).

Let us denote displacements collected over time $\mathbf{t} = \{t - T, \ldots, t - 2, t - 1, t\}$ for any such moving object as $V = \{\mathbf{v}_{t-T}, \mathbf{v}_{t-2}, \mathbf{v}_{t-1}, \ldots, \mathbf{v}_t\}$. A Gaussian process (GP) prior is placed on $f$, such that $f \sim \mathcal{GP}(0, k_{\mathrm{GP}}(\mathbf{t}, \mathbf{t}'))$, and $V = f(\mathbf{t}) + \epsilon$ is an additive noise $\epsilon \sim \mathcal{N}(0, \sigma^2)$. This way we can model non-linear relationships between motion and time. As $\mathbf{v}$ are observations in $2D$, the model is a two dimensional output GP. However, it is also possible to disregard the correlation between response variables $v_x$ and $v_y$ for simplicity. So as to capture the variations in motions, we adopt a polynomial covariance function of degree 3. Further, as commonly used in *kriging* methods in geostatistics [14], we explicitly augment the input with a quadratic term $\tilde{\mathbf{t}} = [\mathbf{t}, \mathbf{t}^2]^\top$ and build $k_{\mathrm{GP}}(\mathbf{t}, \mathbf{t}') = (\tilde{\mathbf{t}}\tilde{\mathbf{t}}' + 1)^3$, to improve (verified in pilot experiments) the prediction. Unlike squared-exponential kernels which definitely decay beyond the range of data points, polynomial kernels are suitable for extrapolation into the near future. However, note that polynomials of unnecessarily higher orders would result in over-fitting.

The predictive distribution for the motion of a point in the locality of an individual GP at a given time, $\mathbf{v}_* \sim \mathcal{N}(\mathbb{E}, \mathbb{V})$, can be then predicted using standard GP prediction equations [15] (Figure 4). Note that hyperparameters of each GP has to be optimized before making any predictions. The associated distribution for the position of a point transformed by $p(\mathbf{v}(\mathbf{x}))$ is then,

$$\mathbf{s} \sim \mathcal{N}(\rho, \Sigma) \sim \mathcal{N}(\mathbb{E}, \mathbb{V}) \sim \mathcal{N}\left(\begin{bmatrix} x \\ y \end{bmatrix} + \mathbb{E}, \mathbb{V}\right) \sim \mathcal{N}\left(\begin{bmatrix} x + \mu_x \\ y + \mu_y \end{bmatrix}, \begin{bmatrix} \sigma_{xx} & \sigma_{xy} \\ \sigma_{yx} & \sigma_{yy} \end{bmatrix}\right) \quad (2)$$

where we used $\mathbf{s}(\mathbf{x})$ to denote the spatial coordinates of $\mathbf{x}$ such that $\mathbf{s}(\mathbf{x}) = (x, y)$.

## 3.3 Feature embedding

With the predicted spatial coordinates for each point $\mathbf{x}$ at time $t_*$, represented as $\mathcal{N}(\rho, \Sigma)$, obtained in the previous step, the HF-STHM (hinged feature STHM) can now be constructed. As there is uncertainty in the motion of a point, this uncertainty needs to be propagated into the map.

Denoting $\mathcal{H}$ for a reproducing kernel Hilbert space (RKHS) of functions $f : \mathcal{S} \to \mathbb{R}$ with a reproducing kernel $k : \mathcal{S} \times \mathcal{S} \to \mathbb{R}$, the mean map $\mu$ from probability space $\mathcal{P}$ into $\mathcal{H}$ is obtained [16] as $\mu : \mathcal{P} \to \mathcal{H}, \quad \mathbb{P} \mapsto \int_{\mathcal{S}} k(\mathbf{s}, \cdot) d\mathbb{P}(\mathbf{s})$. Then, the kernel between two distributions can be written as,

$$k(\mathbb{P}_i, \mathbb{P}_j) = \int \int \langle k(\mathbf{s}_i, \cdot), k(\mathbf{s}_j, \cdot) \rangle_{\mathcal{H}} d\mathbb{P}_i(\mathbf{s}_i) d\mathbb{P}_j(\mathbf{s}_j)$$

$$= \int \int k(\mathbf{s}_i, \mathbf{s}_j) d\mathbb{P}(\mathbf{s}_i) d\mathbb{P}(\mathbf{s}_j) \quad (3)$$

$$= \int \int k(\mathbf{s}_i, \mathbf{s}_j) p(\mathbf{s_i}; \rho_i, \Sigma_i) p(\mathbf{s_j}; \rho_j, \Sigma_j) d\mathbf{s}_i d\mathbf{s}_j,$$

where $\langle \cdot, \cdot \rangle$ denotes the dot product and $\mathbb{P}_i := \mathbb{P}(\mathbf{s}_i) = \mathcal{N}(\rho_i, \Sigma_i)$ in a probability space $\mathcal{P}$.

**Theorem 1** *[17] If a squared exponential kernel, $k(\mathbf{s}_i, \mathbf{s}_j) = \exp\{-\frac{1}{2}(\mathbf{s}_i - \mathbf{s}_j)^\top L^{-1}(\mathbf{s}_i - \mathbf{s}_j)\}$, is endowed with $\mathbb{P} = \mathcal{N}(\mathbf{s}; \rho, \Sigma)$, then there exists an analytical solution in the form,*

$$k(\mathbb{P}_i, \mathbb{P}_j) = \left| I + L^{-1}(\Sigma_i + \Sigma_j) \right|^{-1/2} \exp\left\{ -\frac{1}{2}(\rho_i - \rho_j)^\top (L + \Sigma_i + \Sigma_j)^{-1}(\rho_i - \rho_j) \right\}, \quad (4)$$

*where $I$ is the identity matrix and $L$ is the matrix of length scale parameters which determines how fast the magnitude of the exponential decays with $\rho$.*

**Corollary 1** *For point estimates $\tilde{\mathbf{s}}$ of $\mathbb{P}_j$,*

$$k(\mathbb{P}_i, \tilde{\mathbf{s}}) = \left| I + L^{-1}\Sigma \right|^{-1/2} \exp\left\{ -\frac{1}{2}(\rho - \tilde{\mathbf{s}})^\top (L + \Sigma)^{-1}(\rho - \tilde{\mathbf{s}}) \right\}. \quad (5)$$

Corollary 1 is now used to compute $k(p(\mathbf{s}), \tilde{\mathbf{s}})$ which defines the feature embedding for HF-STHM. Note that Corollary 1 is equivalent to centering (hinging) the kernels at $M$ fixed points $\tilde{\mathbf{s}}$ in space which allows capturing different spatial dependencies over the map dimensions. The pooled-length scales $L + \Sigma$ of these "hinged" kernels change over time. Typically, these $\tilde{\mathbf{s}}$ can be obtained by a pre-defined regular grid. Finally, the feature mapping for each spatial location is obtained by concatenating multiple kernels hinged at supports:

$$\Phi_{\text{hinged}}(\mathbf{x}) = \left[ k(p(\mathbf{s}), \tilde{\mathbf{s}}_1), \ldots, k(p(\mathbf{s}), \tilde{\mathbf{s}}_M) \right]^\top, \quad (6)$$

The method to predict occupancy maps at each iteration is summarized in Algorithm 1. As in SHM, the length-scale of the hinged-feature kernels and the regularization parameter has to be picked heuristically or using grid-search.

**Data:** Set of consecutive laser scans
**Result:** Continuous occupancy map at time $t_*$ at any arbitrary resolution
**while** *true* **do**
  | Extract motion observations $V$ (Section 3.1);
  | Build the motion vector field from $V$ using Gaussian process regression (Section 3.2);
  | Generate motion predictions $p(\mathbf{v})$ for $t_*$ (Section 3.2);
  | Compute the feature mapping (Equation 6);
  | Update $\mathbf{w}$ of the logistic regression model similar to Section 2;
  | Generate a new spatial map by querying at a desirable resolution similar to Section 2;
**end**

<div align="center">

**Algorithm 1:** Querying maps for $t_*$ using HF-STHM algorithm.
</div>

Being a parametric model, this method can be used to predict past ($t_* < 0$), present ($t_* = 0$) and future ($t_* > 0$) occupancy maps using a fixed number of parameters ($M + 1$). However, in practice, it may not be required to generate future or past maps at every time step. However, it is required to incorporate new laser data and update $\mathbf{w}$ using SGD at each iteration. Therefore, GP predictions and probabilistic feature embedding can be skipped by setting $\Sigma = \mathbf{0}$, whenever it is not required to predict future or past maps as the uncertainty of knowing the current location for any laser reflection is zero.

## 4  Experiments and Discussion

In this section we demonstrate how HF-STHM can be effectively used for mapping in dynamic environments. Our main dataset[1], named as dataset 1, consists of laser scans, each with 180 beams covering $180^0$ angle and 30 m radius, collected from a busy intersection [6]. Figure 3 [6] shows an aerial view of the area and the location of the sensor. In Section 4.4, we used an additional dataset[1] (dataset 2) of a larger intersection, as this section verifies an important part of our algorithm.

### 4.1  Motion model

Figure 4 shows a real instance where a vehicle breaks and how the GP model is cable of predicting its future locations with associated uncertainty. Although the GP has two outputs $v_x$ and $v_y$, only predictions along the direction of motion $v_x$ is shown for clarity. There can be several such GP models at a given time as a new GP model is initialized for each new moving object (centroid association) entering the environment and is removed as it disappears. The GP model not only extrapolates the motion into the future, but also provides an estimate of the predictive uncertainty which is crucial for the probabilistic feature embedding techniques discussed in Section 3.3. This location uncertainty around past observations is negligible while it is increasingly high as the more time steps ahead into the future we attempt to predict. However, the variance may also slightly change with the number of data points in the GP and the variability of the motion. As opposed to the two-frame based velocity calculation technique employed in DGPOM, our method uses motion data of dynamic objects collected over several frames which makes the predictions more accurate as it does not make assumptions about the motion of objects such as constant velocity.

### 4.2  Supports for hinged features

Although in Section 3.3 we suggested to hinge the kernels using a regular grid, we compare it with kernels hinged in random locations in this experiment. As shown in Table 1, the area-under-ROC-curve (AUC) averaged over randomly selected maps at $t_* = 0$ are more accurate for regular grid because random supports cannot cover the entire realm, especially if the number of supports is small. Similarly, a random support based map may not be qualitatively appealing. In general, regular grid requires less amount of features to ensure a qualitatively and quantitatively better map.

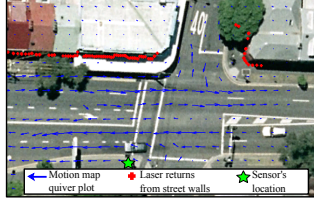

Figure 3: Ariel view of dataset 1 environment

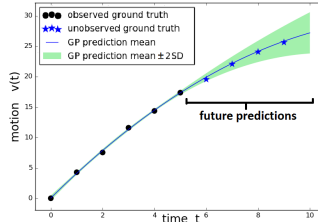

Figure 4: GP model

Table 1: Average AUC – supports for hinged features

| No. of supports | Regular grid | Random grid |
|---|---|---|
| 250 | 0.95 | 0.83 |
| 500 | 0.98 | 0.88 |
| 1000 | 0.99 | 0.94 |
| 5000 | 0.99 | 0.98 |

## 4.3 Point estimate vs. distribution embedding

It is important to understand if distribution embedding discussed in Section 3.3 indeed improves accuracy over point embedding. In order to see this, the accuracy between dynamic clusters of future maps and corresponding ground truth laser values should be compared. Since automatically identifying dynamic clusters is not possible, we semi-automatically extracted them. To this end, dynamic clusters of each predict-ahead map were manually delimited using python graphical user interface tools and negative-log-loss (NLL) between those dynamic clusters and corresponding ground truth laser values were evaluated. Because the maps are probabilistic, NLL is more representative than AUC.

Keeping all other variables unaltered, the average decrements of NLL from point estimates to distribution embedding of randomly selected instances for query time steps $t_* = 1$ to $5$ were $0.11, 0.22, 0.34, 0.83, 0.50, 1.36$ (note! log scale) where $t_* > 0$ represents future. Therefore, embedding both mean and variance, rather than merely mean, is crucial for a higher accuracy. Intuitively, though we can never predict the exact future location of a moving vehicle, it is possible to predict the probability of its presence at different locations in the space.

## 4.4 Spatial maps vs. spatio-temporal maps

In order to showcase the importance of spatio-temporal models (HF-STHM) over spatial models (SHM), NLL values of a subset of dataset were calculated similar to Section 4.3 for compare dynamic occupancy grid map (DGM), SHM and HF-STHM. SHM and HF-STHM used 1000 bases. DGM is an extension to [1] which calculates occupancy probability based on few past time steps. In this experiment we considered 10 past time steps and 1 m grid-cell resolution for DGM.

The experiments were performed for datasets 1 and 2 and results are given in Table 2. The smaller the NLL, the better the accuracy is. HF-STHM outperforms SHM and this effect becomes more prominent for higher $t_*$. DGM struggles in dynamic environments because of the fixed grid-size, assumptions about cell independence and it was not explicitly designed for predicting into the future. NLL of DGM increases with $t_*$ as it keeps memory in a decaying-fashion for 10-consecutive-past-steps. Since SHM does not update positions of objects (as it is a spatial model), NLL also increases with $t_*$. In HF-STHM, NLL increases with $t_*$ because predictive variance increases with $t_*$ in addition to mean error. Figure 5 presents a qualitative comparison.

Table 2: NLL - predictions using dynamic occupancy grid map (DGM), static Hilbert map (SHM) and the proposed method (HF-STHM) for future time steps.

| Time | Dataset 1 | | | Dataset 2 | | |
|---|---|---|---|---|---|---|
| | DGM | SHM | STHM | DGM | SHM | STHM |
| $t_* = 0$ | 11.20 | 0.11 | 0.12 | 6.00 | 0.18 | 0.09 |
| $t_* = 1$ | 17.69 | 0.15 | 0.15 | 10.16 | 0.29 | 0.12 |
| $t_* = 2$ | 19.88 | 0.28 | 0.18 | 12.71 | 0.82 | 0.34 |
| $t_* = 3$ | 25.24 | 0.61 | 0.19 | 16.54 | 1.85 | 0.57 |
| $t_* = 4$ | 26.84 | 1.18 | 0.48 | 20.76 | 2.96 | 0.16 |
| $t_* = 5$ | 27.44 | 1.46 | 0.89 | 25.25 | 4.00 | 1.10 |
| $t_* = 6$ | 34.54 | 2.00 | 1.68 | 26.78 | 4.90 | 1.30 |

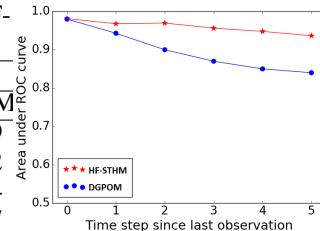

Table 3: AUC of prediction

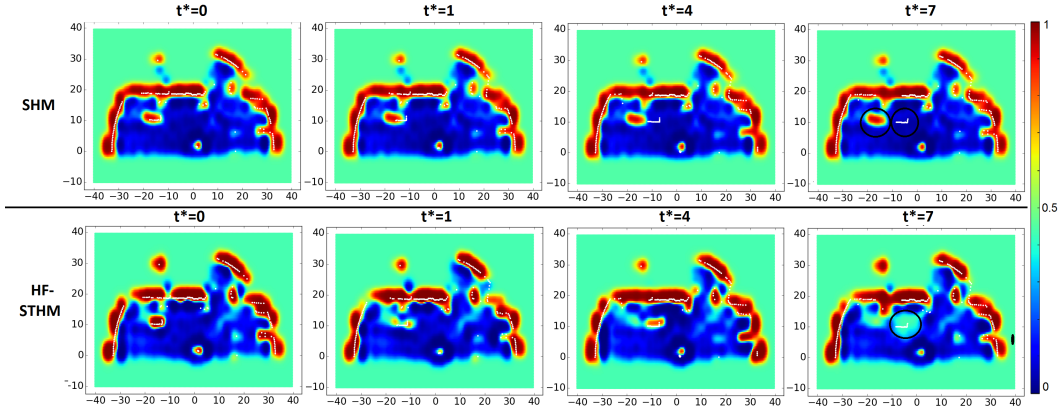

Figure 5: SHM and HF-STHM for $t_*$-ahead predictions. The robot is at (0,0) facing up. The white points are ground truth laser reflections. Observe that, in HF-STHM, moving objects are predicted-ahead and uncertainty of dynamic areas grows as $t_*$ increases. Differences are encircled for $t_* = 7$.

## 4.5 Predicting into the future and retrieving old maps

In order to assess the ability of our method to predict the future locations of dynamic objects, we compare the map obtained when predicting a certain number of time steps ahead ($t_*$) with the measurements made at that time. Then the average is computed and the AUC as a function of how far ahead the model makes predictions. The experiment was carried out similar to [6]We compare our model with DGPOM (AUC values obtained from [6]) as this is the only other method capable of this type of prediction. According to Figure 3 we can see that both methods perform comparably when $t_* < 2$. However, if we predict further ahead our method maintains high quality while DGPOM start to suffer somewhat. One explanation for this is the way motion predictions are integrated in our method. As discussed in Section 4.3, we embed distributions rather than point observations to the model and hence it allows us to better deal with the uncertainty of the motion of the dynamic objects. On the other hand, our motion model can capture non-linear patterns.

In addition to predicting into the future, our method is also capable of extrapolating few steps into the past merely by changing the time index $t$ to negative instead of positive. This allows us to retrieve past maps without having to store the complete dataset. In contrast to DGPOM, the parametric nature and amenability to optimization using SGD makes our method much more efficient in both performing inference and updating with new observations.

## 4.6 Runtime

To add a new observation, i.e. a new laser scan, into the HF-STHM map it takes around $0.5\,\mathrm{s}$ with the extraction of the dynamic objects taking up the majority of the time. To query a single map with $0.1\,\mathrm{m}$ resolution takes around $0.5\,\mathrm{s}$ as well. These numbers are for a simple Python based implementation.

## 5 Conclusions and future work

This paper presented *hinged features* to model occupancy state of dynamic environments, by generalizing static Hilbert maps into dynamic environments. The method requires only a small number of data points (180) per frame to model the occupancy of a dynamic environment (30 meter radius) at any resolution. To this end, uncertainty of motion predictions were embedded into the map in a probabilistic manner by considering spatio-temporal relationships. Because of the hierarchical nature, the proposed feature embedding technique is amenable for more sophisticated motion prediction models and sensor fusion techniques. The power of this method can be used for planning and safe navigation where knowing the future state of the world is always advantageous. Furthermore, it can be used as a general tool for learning behaviors of moving objects and how they interact with the space around them.

## Footnotes

[1]https://goo.gl/f9cTDr

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
