[Supplementary Material · nips-supplementary-ready.pdf]

# Spatio–Temporal Hilbert Maps for Continuous Occupancy Representation in Dynamic Environments - Supplementary Material

**Ransalu Senanayake**
University of Sydney
rsen4557@uni.sydney.edu.au

**Lionel Ott**
University of Sydney
lionel.ott@sydney.edu.au

**Simon O'Callaghan**
Data61/CSIRO, Australia
simon.ocallaghan@data61.csiro.au

**Fabio Ramos**
University of Sydney
fabio.ramos@sydney.edu.au

## 1   Other related work

In this section we discuss alternative approaches to occupancy mapping in dynamic environments and discuss the pros and cons of these in the context of hinged features based spatio-temporal Hilbert maps (HF-STHM). Modeling dynamic environments has been historically divided into two main lines of methods. The first exploits probabilistic sensor models and assumptions about the environment to remove measurements associated with dynamic objects, for example [1]. This results in a more robust static map that is less prone to be corrupted by moving objects.

The second approach augments occupancy grid maps with explicit representations of the dynamics of the environment by modifying the representation of the individual grid cells. [2] propose a method that retains all observations made by the robot by representing them as time intervals with similar perceptions. Another approach, presented by [3], represents each cell as an independent Markov chain with two states. The transition parameters between the two states are learned online and allows the expected dynamics of the cell to be modeled. [4] represent the dynamics of the individual cells with a hidden Markov model which models the transitions between occupied and free states. Another approach is taken by [5] who model each cell as a frequency spectrum, capturing the state of the cell at various time scales over recurring time frames. This allows the map to represent periodic states in the environment at various time scales. These methods are capable of predicting the state and behavior of a single cell, however, they suffer from the independence assumption of grid maps and as such cannot reason about larger patterns in the map. In addition, the above methods are not designed to propagate knowledge from the current state into the future. While [5] captures occupancy at specific frequencies in time this is more intended for regularly recurring events rather then unique events.

To the best of our knowledge there is only one recent method that models occupancy in dynamic environments in a continuous fashion. The dynamic Gaussian processes occupancy maps (DGPOM) [6] builds a velocity vector field of the environment by observing multiple measurements taken consecutively. The vector field output is integrated into a Gaussian process occupancy maps (GPOM) through a covariance function. This approach, similarly to the the HF-STHM, can make occupancy predictions into the future. The main disadvantage is the computational cost incurred by the inversion of a large covariance matrix containing the measurements obtained through space and time. The parametric nature and amenability to optimization using stochastic gradient descent (SGD) makes our method much more efficient in both performing inference and updating with new observations.

## 2 hinged features based spatio-temporal Hilbert maps (HF-STHM)

Another example of HF-STHM is shown in Figure 1. $t = 0$ is the current time frame and a vehicle at the bottom left is moving from center of the image towards the left. The more time steps ahead the model tries to predict ($t_*$), the higher the uncertainty of the position is.

Figure 1: Another example for HF-STHM. The spatially diffused pattern indicates uncertainty of the location.

## 3 Other features for spatio-temporal mapping

Three types of features have been used in static Hilbert Maps (SHMs) [7] — sparse features, random Fourier features (RFF) and Nystrom features. *Hinged features* that we suggest in this paper is similar to sparse features. However, it is also possible to use other features to address other aspects of mapping. For instance, we can use random Fourier features to build long-term maps. We also attempted to minimize regularized expected loss functional in kernel logistic regression,

$$\underset{f \in \mathcal{H}}{\mathrm{argmin}} \quad R(f) := \mathbb{E}_{(x,y) \sim \mathbb{P}(x,y)}[\log\left(1 + \exp(-yf(\mathbf{x}))\right)] + \frac{\nu}{2}\|f\|_{\mathcal{H}}^2 \tag{1}$$

using stochastic functional gradient descent, $f_{t+1}(\cdot) = f_t(\cdot) - \eta_t z k(x_{t+1}, \cdot) + \nu f_t(\cdot)\}$ where $z = \frac{\partial}{\partial f}[\log\left(1 + \exp(-yf(\mathbf{x}))\right)]$, $\eta_t$ is the learning rate at iteration $t$ and $f_t(\cdot) = \sum_{i=1}^{t} \alpha_i k(x, \cdot)$. Rather than storing all past observations for gradient calculations, the recent doubly stochastic functional gradient approach [8] where a seed is re-used to generate PRNs can be used. It guarantees the convergence in the RKHS in rate $\mathcal{O}(1/t)$. Since $\alpha$ naturally preserves qualities of space for sequential time steps, this is essentially a spatio-temporal map. Note that RFF-STHM is non-parametric unlike HF-STHM. Nevertheless, we have effectively exploited the non-parametric behavior in a way that we do not require other underlying models.