[Reviews · NeurIPS 2016]

Reviewer 1

Summary

In this paper, the authors present an approach to generalize static Hilbert maps to dynamic environments. The authors suggest two different features to incorporate the variation in time into the spatial domain: Hinged features and random Fourier features. The approach based on hinged features propagates the observed motion uncertainties into the position of the point in the future. The second approach captures long-term occupancy by the means of doubly stochastic functional gradients. Both methods are evaluated on the example of an intersection where car movements were observed for three minutes. The data is collected using laser scans. The authors performed a detailed evaluation for the hinge features and a small comparison the random Fourier features.

Qualitative Assessment

* Might work with cars but as soon as you have human behavior involved I believe this will not work anymore. Imagine a pedestrian walks towards a parked car. In your approach the pedestrian will walk right through the object. How do you model behavior for traffic lights. In general I believe, it is necessary to have an object classification step in between to learn object specific dynamic models that are not just based on the observed velocity of the object. * Line 26: They do not have to disregard the relationship between time and space. The authors should elaborate what exactly they mean here. * Line 93: What is WOLOG standing for? * Line 94: The claim that the integration of sensor movement is trivial is not correct when you consider motion uncertainties and delays in the robot motion data and image processing. The authors are welcome to present real robot evaluations with moving sensors if they disagree. * Line 116: "a_{ij} is the assignment matrix" -> No, it is a value from that matrix. * Section 3.2: The section is a little bit confusing, as it is not clear up to this point where you get v from. It is also not clear how time is integrated. I assume from the position update of the last frame t-1. Please state it more clearly. * Section 3.2: m(X) -> How do you estimate it. * How are objects entering the scene treated? * Theorem 1: where is the proof? * Experiments: How many cars where observed in those three minutes? * Figure 3: Why is there motion on the roof?

Confidence in this Review

2-Confident (read it all; understood it all reasonably well)


Reviewer 2

Summary

This paper is addressing a very interesting and important problem: how to make occupancy maps in space-time, taking advantage of the continuity of the underlying spaces. The basic approach (which appeared in a previous paper) is very interesting and relatively novel: treat the problem as one of supervised learning, by labeling points in space positive (occupied) or negative (free). Then, given a query point, make a probabilistic prediction of its "class." In this paper, the approach is extended to spatio-temporal maps. The interesting underlying problem is how to capture the relationship between time and space and use it to get the appropriate generalizations, and to propagate uncertainty in objects' dynamics into the occupancy map. Two methods for building such models are presented and evaluated on real robot data.

Qualitative Assessment

This paper addresses a very important question in a really interesting and novel way. The exposition is clear up to a point, the approach seems to be very well founded, and the experiments were done with care and address the important claims of the paper. My only concern, and it is a big one, is that right at the crucial steps in the paper, it becomes very terse, and I was unable to follow exactly what is going on. I understand that the space constraints are significant: I would fairly strongly advocate focusing on one of the two approaches and presenting it in more detail, and possibly compressing some discussion of computing the motion centroids and associating them. The data flow figure of how the motion fields are computed is very helpful. But after that, I was lost: exactly how are the motion fields combined to make an overall predictive model? I just needed a concrete explanation of how, given a new query point in space-time, the predicted occupancy would be generated. Section 4 was also too terse for me to follow. This issue is, again, not so much the specific techniques being discussed as a lack of explanation of the overall learning and prediction framework.

Confidence in this Review

1-Less confident (might not have understood significant parts)


Reviewer 3

Summary

The paper proposes two methods based on spatio-temporal Hilbert maps and generalizes it from static domain to dynamic environments without considering time as a separate dimension. These methods are useful for learning behaviors of moving objects and the way they interact with the environment such as how to predict future locations of a moving vehicle when it breaks with some uncertainties. The first method is based on motion uncertainty propagation into the kernel model, while the second one is based on using doubly stochastic functional gradients to build robust kernel maps. The main contribution of the paper is to incorporate uncertainty of dynamic objects into the kernel map using a hierarchical structure which estimates uncertainties using a Gaussian process regression model with inputs obtained using yet another underlying model. Finally, some experiments are conducted to evaluate the performance of each of these methods in some dynamic environments such as traffic intersections.

Qualitative Assessment

It is not very clear what are the main challenges of incorporating time into dynamic environments in compare with earlier existing results. In particular, it would be good to elaborate further on the main differences of the SHM method and the current work. Is there any way to distinguish which of the proposed models will give a better performance? What is the overall complexity and running time of the proposed methods in terms of parameters of the problem? Although sections 3.1.2 or 4.1 gives some bounds, but it is not clear overall what these bounds would be. The test beds and data sets in section 5 are very limited which make it hard to conclude a fair evaluation for the performance of the proposed methods. Moreover, the paper only provides a short summary of the results without much details which at some cases it is even hard to regenerate them. So it would be good to enrich this section by adding more extensive experimental results.

Confidence in this Review

1-Less confident (might not have understood significant parts)


Reviewer 4

Summary

The paper provides a dynamic extension of spatial hilbert maps. The application is in occupancy mapping in dynamic environments. Two features are introduced the hinged features and the random fourier feature

Qualitative Assessment

The authors work on occupancy grids generated from laser scans and introduce two states of observation -1, and +1. This is surprising, as laser scanner data has, due to occlusions, a third 'unobserved' state (e.g. 0). The proposed model uses a GP to model the motion observations. The approach is a minor improvement over existing works, however, the experiments show some improvement. The data of their experiments is unpublished and results are irreproducible, some link to their ROS bag files and algorithmic implementation could improve their presentation. language: p.4 l. 165 'the the' p.6 l. 177 'the method' p.6 l. 180 'for for p.8 l. 257 table 5.2.1 does not exist p.8 Fig. 6 What is shown in the table?

Confidence in this Review

2-Confident (read it all; understood it all reasonably well)


Reviewer 5

Summary

The authors propose a method of extending static Hilbert maps to incorporate temporal features into the spatial domain. Using primarily LIDAR data, they predict motion of objects using Gaussian process regression and a Bayesian approach, learning a mapping of location and time to velocity. This allows for the previous work in static Hilbert maps to be modified and applied to many difficult scenarios, such as autonomous driving, where many dynamic objects and obstacles exist.

Qualitative Assessment

While there are some formatting issues that need to be addressed (particularly lines 116-118 and Tables 1 and 2), this paper is presented very clearly and provides a novel and important approach for handling current issues in relevant robotics fields. In the final version I would like to see better formatting for some of the tables and figures, to make them clearer and to better punctuate the results. Also, more information and breakdown of the runtime would be desirable, as such an algorithm would be most beneficial if it can operate in real-time driving scenarios.

Confidence in this Review

2-Confident (read it all; understood it all reasonably well)


Reviewer 6

Summary

The paper presents an approach to maintain and update a spatio-temporal occupancy map. Given a spatial position and time, the map gives a probability that location is occupied. Their first approach uses pre-processing of laser scans to get observations of motion for objects and fit a GP per object to predict its future motion. These GPs are then used to define features for a logistic regression classifier that predicts the probability a location is occupied. They also propose an approach that uses random Fourier features. They provide experiments to verify the utility of both of these approaches with respect to prior art.

Qualitative Assessment

Overall, the paper was well-written and proposes an improvement on an important problem. As a general point, I'm not sure about the utility of including the Fourier features approach. Right now, that part of the paper seems under motivated and insufficiently evaluated. The best motivation is at the end of section 4, I think it would be better to move this further forward in the paper. If you decide to include this part of the paper, the experiments should be expanded. 5.2.1 currently shows that RFF features perform better in the temporal domain, but does not give a case for why RFF features are good for learning general motion dynamics or priors. I think the explanation of the approach is section 3 is fairly approachable but it isn't until the end of the section that the reader gets a sense of what is being conveyed. I would include a simple example or step through of the process at the beginning of the section to give a broad strokes overview of the approach. It could also be useful to move Figure 1 higher up in the paper and refer to it earlier. I think it is also important to give some intuition for why the approach uses a feature embedding (rather than directly putting a GP on the occupancy map). It is done for efficiency reasons, but this is not made explicit for the reader. Finally, the experiments section could use some work to make them more accessible. There are 6 experiments to evaluate the HF-STHM, please make it easier for readers to determine which figures and tables correspond to which experiments. Figure 2 shows the results of experiment 5.1.4, Figure 3 shows an illustration of the environment, Figure 4 shows the results of experiment 5.1.1, Figure 5 is not mentioned in the text (but I guess that it is the results of experiment 5.1.5). Please try to place figures in the order the appear in the text and include informative captions that explain the significance of the image. Please also make sure that each experiment clearly states: 1) the methods compared; 2) the dependent variables measured; and 3) the results of the experiment. Figure 1 (b) also has some errors in it and should be revised.

Confidence in this Review

2-Confident (read it all; understood it all reasonably well)